# CXR-Seg: A Novel Deep Learning Network for Lung Segmentation from Chest X-Ray Images

**DOI:** 10.3390/bioengineering12020167

**Published:** 2025-02-10

**Authors:** Sadia Din, Muhammad Shoaib, Erchin Serpedin

**Affiliations:** 1Electrical and Computer Engineering Program, Texas A&M University, Doha 23874, Qatar; 2Department of Electrical and Computer Engineering, Abbottabad Campus, COMSATS University Islamabad, Abbottabad 22060, Pakistan; 3Department of Electrical and Computer Engineering, Texas A&M University, College Station, TX 77840, USA

**Keywords:** lung segmentation, CXR image segmentation, convolutional neural networks, deep learning

## Abstract

Over the past decade, deep learning techniques, particularly neural networks, have become essential in medical imaging for tasks like image detection, classification, and segmentation. These methods have greatly enhanced diagnostic accuracy, enabling quicker identification and more effective treatments. In chest X-ray analysis, however, challenges remain in accurately segmenting and classifying organs such as the lungs, heart, diaphragm, sternum, and clavicles, as well as detecting abnormalities in the thoracic cavity. Despite progress, these issues highlight the need for improved approaches to overcome segmentation difficulties and enhance diagnostic reliability. In this context, we propose a novel architecture named CXR-Seg, tailored for semantic segmentation of lungs from chest X-ray images. The proposed network mainly consists of four components, including a pre-trained EfficientNet as an encoder to extract feature encodings, a spatial enhancement module embedded in the skip connection to promote the adjacent feature fusion, a transformer attention module at the bottleneck layer, and a multi-scale feature fusion block at the decoder. The performance of the proposed CRX-Seg was evaluated on four publicly available datasets (MC, Darwin, and Shenzhen for chest X-rays, and TCIA for brain flair segmentation from MRI images). The proposed method achieved a Jaccard index, Dice coefficient, accuracy, sensitivity, and specificity of 95.63%, 97.76%, 98.77%, 98.00%, and 99.05%on MC; 91.66%, 95.62%, 96.35%, 95.53%, and 96.94% on V7 Darwin COVID-19; and 92.97%, 96.32%, 96.69%, 96.01%, and 97.40% on the Shenzhen Tuberculosis CXR Dataset, respectively. Conclusively, the proposed network offers improved performance in comparison with state-of-the-art methods, and better generalization for the semantic segmentation of lungs from chest X-ray images.

## 1. Introduction

Among the current medical imaging techniques, the chest X-ray (CXR) image is the most often used diagnosis tool because of its affordability, accessibility, and ease of acquisition [1]. Multiple anomalies can be detected simultaneously in a single CXR image. Radiologists must manually validate them. However, radiologists in particular face a difficult challenge when it comes to analysing a large number of CXRs. It takes time, which slows down the diagnosis of patients [2]. An automated CXR image analysis system’s foundation is the accurate segmentation of the lungs in CXR images since the lungs are the region of interest for numerous pulmonary and thoracic conditions, including lung cancer, emphysema, cardiomegaly, and tuberculosis. Additionally, lung contours can provide crucial and valuable information for the detection of diseases that pose a threat to life [3]. Using deep learning models, lung segmentation has demonstrated remarkable performance in segmenting the region of the lungs from additional organs [4]. However, lung segmentation poses several challenges: (1) Non-pathological variations: lung shape and size vary with factors such as age, gender, and heart size. (2) Pathological changes: severe lung diseases can cause opacities with high-intensity values. (3) Obstructions: foreign bodies and medical equipment can obscure the lung field in X-ray images.

To address the above challenges, in this study we consider a deep learning approach for segmenting lung regions in chest X-ray images. The proposed network leverages a pre-trained EfficientNetV2S1 model to extract initial features. A spatial enhancement module refines these features by promoting the integration of spatial information across neighbouring regions. Furthermore, a transformer attention module focuses on extracting intricate local details crucial for lung segmentation. Finally, a multi-scale feature block progressively decodes the refined features to generate a detailed prediction map that precisely delineates the lung boundaries within the chest X-ray image. By incorporating these components into the network architecture, the proposed model aims to achieve robust and accurate lung segmentation from CXR images.

The main constituent modules of the proposed CXR-Seg Network are as follows:1.Pre-trained EfficientNet encoder: EfficientNetV2S1 is utilized to extract feature encodings from the input CXR images.2.Spatial enhancement module (SEM): The SEM is integrated within the skip connection to facilitate the fusion of adjacent features. This module enhances the spatial information within the feature maps, promoting better feature representation for lung segmentation.3.Transformer attention module (TAM): Positioned at the bottleneck layer of the network, the TAM leverages transformer-based attention mechanisms to extract rich local features and refines the extracted feature information. The refinement step aids in improving the accuracy of lung segmentation from CXR images.4.Multi-scale feature fusion block (MS-FFB) at the decoder: This is implemented to perform feature decoding layer by layer more effectively. This block assists in generating a dense prediction map by incorporating multi-scale information into the decoding process.

The rest of this paper is organized as follows. Section 2 discusses related works from the field of lung semantic segmentation in CXR images. Section 3 describes the architecture of the proposed method in detail, including the network design, the components used, and the rationale behind their incorporation. Section 4 presents the computer simulation results obtained by evaluating the proposed method on various benchmark datasets. Performance comparisons with prior studies are conducted to assess the effectiveness and superiority of the proposed method. Section 6 concludes this study by summarizing the key findings, highlighting the merits of the proposed method, and discussing potential directions for future research.

## 2. Related Work

### 2.1. Encoder–Decoder Architectures

The field of biomedical imaging processing has witnessed significant advancements, with deep learning models emerging as the preferred technique for image segmentation tasks [5,6]. In the context of lung cancer segmentation, Hwang et al. [7] proposed a deep CNN architecture that employs atrous convolutions for detecting and segmenting lung regions. Islam et al. [8] proposed a lung segmentation method based on the encoder–decoder architecture U-Net. Liu et al. [9] introduced an enhanced U-Net for accurate lung zone detection and segmentation, utilizing a pre-trained EfficientNet-B4 as an encoder in their model. A U-Net architecture to segment lung regions in chest X-rays was employed by Chen et al. [10]. The proposed multi-label classifier outputs prediction scores associated with various lung pathologies, including cardiomegaly, emphysema, and lung nodules [10]. The performance of the multi-label classifier was evaluated on three publicly available CXR datasets: MC, JSRT, and NIH Chest X-ray. The U-Net architecture showed improved segmentation of CXR images relative to the state-of-the-art methods  [10]. Maity et al. [11] introduced a deep convolutional neural network (DCNN) specifically designed for semantic segmentation of lungs in chest X-rays, captured from either a posteroanterior or anteroposterior view [11]. Their DCNN architecture builds upon UNet++ [12], leveraging EfficientNet-B4 as its backbone and incorporating residual blocks within the decoder to mitigate performance degradation and achieve better results with fewer parameters [11]. To further enhance performance, preprocessing techniques like the Top-Bottom-Hat transformation and CLAHE, along with data augmentation strategies were employed in [11]. Vidal et al. [13] addressed the challenge of segmenting lung regions in chest X-rays acquired with portable X-ray devices, particularly for lungs affected by COVID-19 [13]. This approach incorporated a U-Net architecture for lung segmentation and employed a two-stage transfer learning process. This approach achieved a high segmentation accuracy (97.61%) for COVID-19 patients on two separate COVID-19 datasets [13]. Dai et al. [14] introduced SCAN (Structure Correcting Adversarial Network), a novel segmentation network consisting of a critic network and a fully convolutional network (FCN) [15]. The FCN generates an initial predicted mask, which is further refined by the critic network to closely resemble the ground truth segmentation. Khan et al. [16] introduced an Expand-Squeeze Dual Multiscale Residual Network (ESDMR-Net), designed specifically for resource-constrained computing hardware like mobile devices. ESDMR-Net focuses on extracting multi-scale features to facilitate learning contextual dependencies among semantically distinct features. Frid-Adar et al. [17] tested the performance of various approaches for segmenting anatomical structures from chest radiographs such as the U-Net model, FCN, Dilated Residual Networks, and Fully Convolutional DenseNet. It turns out that the U-Net equipped with the ImageNet pre-trained encoder outperforms the state-of-the-art approaches for segmentation of anatomical structures from chest radiographs [17].

### 2.2. Attention Mechanisms

Ahila et al. [18] proposed E-GCS network to differentiate between normal and abnormal cases of COVID-19 using three distinct medical imaging modalities: ultrasound imaging, X-ray images, and CT scan images. This is achieved through the implementation of an attention bottleneck residual network (AB-ResNet). Pal et al. [19] introduced attention UW-Net, a novel technique aimed at enhancing the segmentation performance of small lesion patches. This method utilizes skip connections to connect the encoder and decoder in the encoder–decoder architecture, referred to as UW-Net, which comprises densely connected convolutional layers. Tang et al. [20] proposed XLSor (X-ray Lung Segmentor), a novel architecture incorporating an anomalous CXR pair creation method for data augmentation and a Criss-Cross Attention module. Singh et al. [21] developed Deep LF-Net, a semantic lung segmentation approach based on the DeepLabv3+ architecture, using MobileNetV2 as the backbone.

### 2.3. Hybrid Approaches

Liu et al. [22] introduced residual connections and a modified leaky activation function to enhance performance. Their model’s decoder incorporated a block, as outlined in He et al. [23]. This model showcased robustness in accurately segmenting lung regions, even when faced with challenges such as unclear chest X-ray (CXR) images and cases involving abnormalities like pleural effusion and lung deformation. To improve the handling of edge and detail features of lesions, Fu et al. [24] devised the Deep Supervision Feature Refinement Attention Network (DSFRA-Net) through extensive experimentation. DSFRA-Net incorporates several key components for enhanced performance. Selvan et al. [25] proposed a new approach for segmenting lungs in chest X-rays, specifically for those presenting high opacity [25]. Their technique leverages an encoder–decoder architecture that incorporates variational autoencoders (VAEs) [25]. This method offers a unique solution for handling the challenges presented by high-opacity CXRs. LF-SegNet [26] proposed a fully convolutional encoder–decoder architecture for the identification and segmentation of lungs from CXR images. Novikov et al. [27] investigated the use of three fully convolutional structures for detecting and segmenting anatomical structures such as lungs, hearts, and clavicles in chest X-rays  [27]. Their evaluation employed the Jaccard score on 247 CXR images from the JSRT dataset, and InvertedNet emerged as the best-performing method, even exceeding the performance of U-Net and human observers. In a separate study, Gómez et al. [28] focused on the challenge of segmenting multiple organs in CXR images, including hearts, lungs, and clavicles. These studies demonstrate the potential of deep learning techniques for automating medical image analysis tasks in chest X-rays. The success in segmenting lungs from portable devices highlights the potential of this technique in resource-limited settings.

## 3. Proposed Method

An overview of the proposed network for lung segmentation from CXR images is illustrated in Figure 1. The proposed network consists of four major components: (1) a pre-trained EfficientNetV2S1 [29] model to extract diverse features encodings; (2) a spatial enhancement module (SEM) embedded in the skip connection to promote the adjacent features fusion; (3) a transformer attention module (TAM) at the bottleneck layer to extract rich local features and to refine the extracted feature information for lung segmentation from CXR images; and (4) a multi-scale feature fusion block (MS-FFB) at the decoder to enhance the layer-by-layer feature decoding, resulting in the generation of a dense prediction map.

The RGB image $\chi \in R^{H\times W\times 3}$ is input to the proposed network. The output of the first encoder block is denoted by $s_{1}\in R^{H\times W\times C}$ and expressed as in Equation (Equation 1):(1)$$s_{1}=\mathrm{Enc}.{\mathrm{Block}}_{1}\left(\chi \right).$$

Module $\mathrm{Enc}.{\mathrm{Block}}_{1}$ is the first encoder block of the EfficientNetV2S1 [29]. The output of the second encoder block is represented by $s_{2}\in R^{\frac{H}{2}\times \frac{W}{2}\times 2C}$ and determined by applying the 1×1 standard convolution operation on $s_{1}$ followed by the $\mathrm{Enc}.{\mathrm{Block}}_{2}$, as shown in Equation (Equation 2):(2)$$s_{2}=\mathrm{Enc}.{\mathrm{Block}}_{2}\left(f^{1\times 1}\left(s_{1}\right)\right).$$

The notation $f^{1\times 1}$ stands for the standard 1×1 convolution operation. The output of the third encoder block is denoted by $s_{3}\in R^{\frac{H}{4}\times \frac{W}{4}\times 4C}$ and is evaluated by applying a 1×1 convolution operation to $s_{2}$ followed by the $\mathrm{Enc}.{\mathrm{Block}}_{3}$, as illustrated in Equation (Equation 3):(3)$$s_{3}=\mathrm{Enc}.{\mathrm{Block}}_{3}\left(f^{1\times 1}\left(s_{2}\right)\right).$$

The final output of the encoder side $\chi_{out}^{enc}$ is computed by employing the 1×1 convolution operation on $s_{3}$ and then the extracted information is further refined by employing a transformer attention module (TAM). The TAM not only extracts the rich local features but also captures the important global context information for lung region segmentation from CXR images. This operation is formally expressed as in Equation (Equation 4):(4)$$\chi_{out}^{enc}=\mathrm{TAM}\left(f^{1\times 1}\left(s_{3}\right)\right).$$

Once the feature information is extracted from the encoder stage and refined at the bottleneck layer, it is then fed to the decoder stage to fuse and reconstruct the feature information. The output of the last decoder block is denoted by $F_{3}^{dec}$ and is computed by applying MS-FFB to $\chi_{out}^{enc}$ and then concatenating with the output of the spatial enhancement module (SEM) applied to $s_{3}$, as shown in Equation (Equation 5):(5)$$F_{3}^{dec}=\mathrm{SEM}\left(s_{3}\right)⊗\mathrm{MS}-\mathrm{FFB}\left(\chi_{out}^{enc}\right),$$
where ⊗ stands for the concatenation operation. The output of the second decoder block is denoted by $F_{2}^{dec}$ and computed by applying an upsampling operation followed by a 1×1 standard convolution operation and MS-FFB to $F_{3}^{dec}$. Then, it is concatenated with the output of the spatial enhancement module (SEM) applied to $s_{2}$, as illustrated in Equation (Equation 6):(6)$$F_{2}^{dec}=\mathrm{SEM}\left(s_{2}\right)⊗\mathrm{MS}-\mathrm{FFB}\left(f^{1\times 1}\left(M_{up}\left(F_{3}^{dec}\right)\right)\right),$$
where $M_{up}$ represents the max un-pooling operation used for the upsampling of the feature maps. The output of the first decoder block is denoted by $F_{1}^{dec}$ and is computed by applying the upsampling operation followed by the 1×1 standard convolution operation and MS-FFB to $F_{2}^{dec}$ and then concatenating with the output of the spatial enhancement module (SEM) applied to $s_{1}$, as shown in Equation (Equation 7):(7)$$F_{1}^{dec}=\mathrm{SEM}\left(s_{1}\right)⊗\mathrm{MS}-\mathrm{FFB}\left(f^{1\times 1}\left(M_{up}\left(F_{2}^{dec}\right)\right)\right).$$

Finally, the pixel-level prediction results are computed by applying an upsampling operation to $F_{1}^{dec}$ followed by the 1×1 standard convolution and sigmoid operation, as illustrated in Equation (Equation 8):(8)$$X_{out}=\sigma \left(f^{1\times 1}\left(M_{up}\left(F_{1}^{dec}\right)\right)\right),$$
where σ denotes the sigmoid operation and $X_{out}$ represents the final predicted binary mask.

### 3.1. Spatial Enhancement Module (SEM)

Pooling operations are vital for reducing model complexity and computational overhead, and enhancing feature resilience. Max pooling selectively preserves dominant feature responses, capturing nonlinear information, while average pooling retains broader, low-frequency features, offering valuable global context. To integrate both local and global features effectively, we propose employing parallel paths of max-pooling and average-pooling operations. In our approach, depicted in Figure 2, the input feature map undergoes initial processing through convolutional layers, followed by batch normalization and Rectified Linear Unit (ReLU) activation, as shown in Equation (Equation 9):(9)$$i_{1}=ℜ\left(\beta_{n}\left(f^{3\times 3}\left(F_{in}\right)\right)\right).$$

Subsequently, the outputs of the max pooling and average pooling are concatenated, and then subject to the following operations: convolution, batch normalization ($\beta_{n}$), and ReLU activation (*ℜ*), as illustrated in Equation (Equation 10):(10)$$i_{2}=ℜ\left(\beta_{n}\left(f^{3\times 3}\left[P_{\mathrm{Avg}.}^{3\times 3}\left(i_{1}\right)⊗P_{\mathrm{Max}.}^{3\times 3}\left(i_{1}\right)\right]\right)\right).$$

In addition, we introduce an additional pathway incorporating global average pooling ($P^{\mathrm{Global}}\mathrm{Avg}.$), convolutional operations, batch normalization, and sigmoid activation (σ) to generate attention coefficients for weighting the results of parallel pooling. Subsequently, the weighted feature map is combined with the input to produce output $F_{out}$ of the spatial enhancement module (SEM), as depicted in Equation (Equation 11):(11)$$F_{out}=\left[\sigma \left(\beta_{n}\left(f^{3\times 3}\left(P_{\mathrm{Avg}.}^{\mathrm{Global}}\left(F_{in}\right)\right)\right)\right)⊙i_{2}\right]⊕F_{in}.$$

### 3.2. Multi-Scale Feature Fusion Block (MS-FFB)

On the decoder side, we design a multi-scale feature fusion block (MS-FFB; Figure 3) to fuse the refined local features and global context information extracted from the encoder. The proposed MS-FFB conducts the following operations: the standard 2-D convolution followed by the activation function, batch normalization, depth-wise separable convolution, activation function, and batch normalization. The output of the proposed MS-FFB is computed as shown in Equation (Equation 12):(12)$$\mathrm{MS}-\mathrm{FFB}=\sum_{i=1}^{3}\left(\beta_{n}\left(ℜ\left(f_{\mathrm{DWS}}^{k\times k}\left(\beta_{n}\left(ℜ\left(f^{k\times k}\left(X_{in}\right)\right)\right)\right)\right)\right)\right),$$
where k=2i−1, $X_{in}$ is the input to the MS-FFB, *ℜ* stands for the ReLU activation function, $\beta_{n}$ denotes the batch normalization layer, and $f^{k\times k}$ and $f_{\mathrm{DWS}}^{k\times k}$ represent the k×k standard and depth-wise separable convolution operations.

### 3.3. Transformer Attention Module (TAM)

Given that multi-head attention can learn self-correlation but cannot learn spatial information, a commonly used approach in academic works is to pass the feature map to a position encoding block and then input it to the multi-head transformer self-attention block, as illustrated in Figure 4. The input feature map $F_{in}$ is then embedded in three matrices, $Q\in R^{(h\times w)\times c}$, $K\in R^{c\times (h\times w)},$ and $V\in R^{c\times (h\times w)}$, via the following maps:(13)$$Q=W_{Q}\cdot F_{in}$$(14)$$K=W_{K}\cdot F_{in}$$(15)$$V=W_{V}\cdot F_{in},$$

$W_{Q}$, $W_{K}$, and $W_{V}$ denote three distinct embedding functions responsible for linear projections. The normalized scaled dot product, followed by the softmax operation, is applied between *Q* and *K*, which signifies the similarity between channels in *Q* and others. To compute the aggregated values weighted by attention scores, the spatial contextual attention map is multiplied by the value matrix *V*. Consequently, the expression for the multi-head transformer self-attention mechanism can be represented as(16)$$A_{tsa}\left(Q,K,V\right)=Softmax\frac{QK}{\sqrt{d_{k}}}V.$$
where $d_{k}$ is the scaling factor. Finally, $A_{tsa}$, where $d_{a}\in R^{c\times (h\times w)}$, is reshaped to $R^{h\times w\times c}$, resulting in a tensor of the same size as the input.

### 3.4. Evaluation Matrices

The performance evaluation of the proposed network is carried out using five standard metrics: accuracy, Jaccard index (IOU), Dice coefficient, sensitivity, and specificity. The performance measures are calculated based on predictions of true positives ($T_{P}$), true negatives ($T_{N}$), false positives ($F_{P}$), and false negatives ($F_{N}$), as illustrated in Equations (Equation 17)–(21).(17)$$\begin{array}{ccc} \mathrm{Accuracy}\;(A_{\mathrm{cc}})\; & = & \;\frac{T_{P}+T_{N}}{T_{P}+T_{N}+F_{P}+F_{N}} \end{array}$$(18)$$\begin{array}{ccc} \mathrm{Sensitivity}\;(S_{n})\; & = & \;\frac{T_{P}}{T_{P}+F_{N}} \end{array}$$(19)$$\begin{array}{ccc} \mathrm{Jacard}\;\;\mathrm{Index}\;(J)\; & = & \;\frac{T_{P}}{T_{P}+F_{P}+F_{N}} \end{array}$$(20)$$\begin{array}{ccc} \mathrm{Dice}-\mathrm{Score}\;(D)\; & = & \;\frac{2T_{P}}{2T_{P}+F_{P}+F_{N}} \end{array}$$(21)$$\begin{array}{ccc} \mathrm{Specificity}\;(S_{p})\; & = & \;\frac{T_{N}}{T_{N}+F_{P}}. \end{array}$$

### 3.5. Training Details

During model training, the images from the datasets underwent augmentation, including contrast adjustments (scaling factors of [×0.9,×1.1]) and horizontal and vertical flipping operations, effectively increasing the dataset size by a factor of 5. We employed the Adam optimizer with a maximum of 60 iterations and an initial learning rate set to 0.001. To mitigate overfitting, a strategy of reducing the learning rate by a quarter was adopted if there was no performance improvement on the validation set after five epochs. Additionally, an early-stop strategy was implemented. The models were developed using Keras with TensorFlow as the backend and trained on an NVIDIA K80 GPU.

## 4. Experiments

### 4.1. Datasets

We conducted experiments using three datasets that are widely used for lung segmentation (MC, Darwin, and Shenzhen) and brain tumour segmentation (TCIA). The description of the datasets is given in Table 1 and summarized as follows.

#### 4.1.1. MC Dataset [30]

The MC dataset [30] originates from the Department of Health and Human Services of the Montgomery County in the United States. It comprises 138 CXR images, each with a resolution of 4892×4020 or 4020×4892 pixels, along with their respective masks. The annotations were performed under the radiologist’s supervision. Specifically, the dataset contains 58 images depicting tuberculosis disease and 80 normal images.

#### 4.1.2. Darwin COVID-19 CXR Images Dataset [31]

The Darwin Lung Segmentation Dataset [31] comprises chest X-ray images annotated specifically for lung segmentation purposes. It serves as a fundamental resource in medical image analysis research, facilitating the development and assessment of algorithms designed for automatic lung segmentation in chest X-ray images. The dataset contains 6500 CXR images each with a resolution of 5600×4700 to 157×157 pixels, accompanied by pixel-level polygonal lung segmentations binary masks. These masks were meticulously annotated by radiologists. There are 517 cases of COVID-19 among these images.

#### 4.1.3. Shenzhen Tuberculosis CXR Dataset [30]

The Shenzhen Tuberculosis CXR Dataset [30] is a publi cly available dataset compiled by Shenzhen No. 3 People’s Hospital and Guangdong Medical College in China in 2012. This dataset comprises 662 CXR images, each with a resolution of 3000×3000 pixels, accompanied by binary masks. These masks were meticulously annotated by radiologists. The dataset encompasses 336 images depicting tuberculosis disease and 326 normal images.

#### 4.1.4. TCIA Dataset [32]

The TCIA [32] brain tumour dataset consists of 1084 images, each resized to a uniform 256 × 256 pixels for standardization. The dataset is divided into 80% for training (868 images), 10% for validation (108 images), and 10% for testing (108 images). To enhance the training set, data augmentation is applied, expanding it to 4340 images, while the validation set is increased to 540 images. No augmentation is applied to the test set. This dataset is specifically utilized for brain tumour segmentation tasks.

### 4.2. Ablation Experiments

#### 4.2.1. Ablation on Network Components

An ablation study of the proposed CXR-Seg Network was performed on the MC dataset [30]. Experiments were conducted progressively. We started with the implementation of the baseline encoder–decoder network followed by its multi-scale implementation. The multi-scale module improves the network’s ability to detect features at various resolutions, allowing it to better handle variations in lesion size, shape, and texture. After that, we employed the TAM module in the bottleneck layer of the proposed CXR-Seg Network. The transformer attention module enhances the network’s awareness of global context by recording long-range pixel associations, which aid in the perception of spatial patterns, particularly in complicated locations. Then, we employed the SEM module on the skip connections between encoder–decoder blocks of the proposed CXR-Seg Network. The SEM improves spatial detail by focusing on essential areas and decreasing noise, resulting in more accurate segmentation. Finally, we combined all the modules to perform the segmentation task accurately. When these modules are combined, they significantly increase the network’s segmentation ability, as evidenced by the ablation study, resulting in higher accuracy and better boundary identification across all measures. Figure 5 presents a visual analysis of the ablation experiments performed on the different network components of the proposed CXR-Seg Network. The quantitative outcomes of the ablation study are presented in Table 2. The integration of various components into the proposed CXR-Seg Network led to a notable enhancement in the segmentation performance. Table 3 presents the quantitative results of the ablation study performed on different backbones for the proposed CXR-Seg Network. The results of Table 3 demonstrate that EfficientNet achieves the highest performance, making it the optimal choice for the backbone of the CXR-Seg Network.

#### 4.2.2. Performance Evolution Versus Epoch

The performance evolution with respect to the number of epochs and comparisons of the proposed CXR-Seg Network with other methods are presented in Figure 6. Throughout the training of the various networks, we track the evolution of the Jaccard index and loss function as the number of epochs progresses. It can be observed from Figure 6 that the Jaccard index of the proposed CXR-Seg Network crosses 80% in the 15th epoch, whereas Swin-UNet [33] crosses 80% in the 20th epoch. The maximum Jaccard index of the proposed CXR-Seg Network goes up to 95.63%, whereas the second best method, i.e., U-Net++ [12], achieves the maximum Jaccard index of 94.57%, which is 1.6% less than the proposed CXR-Seg Network. Similarly, the proposed CXR-Seg Network converges faster than the other methods and achieves the minimum loss compared to all other methods.

## 5. Results and Discussion

We compared the performance of the CXR-Seg Network with respect to the other publicly available state-of-the-art models, including Attention Res-UNet [34], BCDU-Net [35], DUCK-Net [36], Swin-UNet [33], U-Net [37], and UNet ++ [12]. We trained all of these models and subsequently tested their performance.

The evaluation of the CXR-Seg Network for chest X-ray image segmentation on the MC dataset (refer to Table 4) demonstrates the superior performance of our proposed method across all evaluation metrics. Specifically, in comparison to the state-of-the-art (SOTA) methods listed in the literature, the CXR-Seg Network outperforms them by margins ranging from 0.98% to 4.93% in terms of the Jaccard index for the MC dataset. This superiority is corroborated through visual inspection (refer to Figure 7), illustrating that the output generated by the CXR-Seg Network closely aligns with the ground truth data. Notably, our method exhibits robustness in handling images with varying sizes and low contrast, further reinforcing its efficacy in clinical chest X-ray image segmentation tasks.

Upon scrutinizing the performance of the CXR-Seg Network for chest X-ray image segmentation on the V7 Darwin COVID-19 Chest X-ray Dataset (refer to Table 5), it becomes apparent that our proposed method outperforms competing approaches across all evaluation metrics. Specifically, in comparison to SOTA methods listed in the literature, the CXR-Seg Network achieves higher scores, ranging from 1.41% to 2.24%, in terms of the Jaccard index on the V7 Darwin COVID-19 Chest X-ray Dataset. This superiority is visually evident in Figure 8, where comparisons with other recent methods show the superior performance of the CXR-Seg Network. The visual results of our method notably surpass those of all other methods on the V7 Darwin COVID-19 Chest X-ray Dataset [31].

Our analysis of the performance of the CXR-Seg Network for chest X-ray image segmentation on the Shenzhen Tuberculosis CXR Dataset (see Table 6) reveals its superiority across all evaluation metrics. Particularly noteworthy is our method’s superior performance relative to the listed SOTA methods, by margins ranging from 0.11% to 4.75%, in terms of the Jaccard index on the Shenzhen Tuberculosis CXR Dataset. This superiority is visually reinforced through the comparisons presented in Figure 9, which vividly depict the improved visual results achieved by the CXR-Seg Network over recent methods on the Shenzhen Tuberculosis CXR Dataset [30].

Table 7 illustrates a statistical comparison between the proposed CXR-Seg model and several SOTA methods concerning brain flair segmentation. This comparison is conducted using the TCIA dataset [32], which presents several challenges such as varying sizes, irregular shapes, and low contrast. Compared to Attention Res-UNet [34], BCDU-Net [35], DUCK-Net, SwinU-Net [33], U-Net [37], and UNet++ [38], the Jaccard index of CXR-Seg exhibits improvements of 7.87%, 18.47%, 10.39%, 13.77%, and 11.35%, respectively, for flair segmentation of the brain. Figure 10 provides visual representations of the brain flair segmentation results. Notably, the proposed CXR-Seg achieves the most accurate segmentation results, closely resembling the ground truth data, even for images characterized by varying sizes, irregular shapes, and low contrast.

In Table 8, we present the computational cost analysis of the proposed CXR-Seg Network with the other methods. It is evident from Table 8 that the proposed CXR-Seg Network has a very small number of learnable parameters, requires very little memory, and has reduced inference time in comparison with the other methods, which makes CXR-Seg Network a suitable choice for clinical application due to its lightweight nature, diagnostic accuracy, improvement in workflow efficiency, and support of early disease detection.

## 6. Conclusions

This study proposed a deep learning approach for segmenting lung regions in chest X-ray images. The proposed network utilizes the pre-trained EfficientNetV2S1 model for initial feature extraction, followed by a spatial enhancement module to integrate spatial information across neighbouring regions. A transformer attention module captures intricate local details essential for accurate lung segmentation, while a multi-scale feature block refines and decodes features to produce precise lung boundary delineations in chest X-ray images. The proposed model demonstrated superior performance compared to other state-of-the-art models. Future directions include using transformer attention modules for 3D medical imaging applications, evaluating model generalization on diverse datasets, addressing lung segmentation challenges, and expanding tasks to include other medical conditions. Addressing biases in chest X-rays, augmentation, domain adaption, fairness indicators, and regular updates is crucial for accurate and equal outcomes.

## Figures and Tables

**Figure 1 bioengineering-12-00167-f001:**
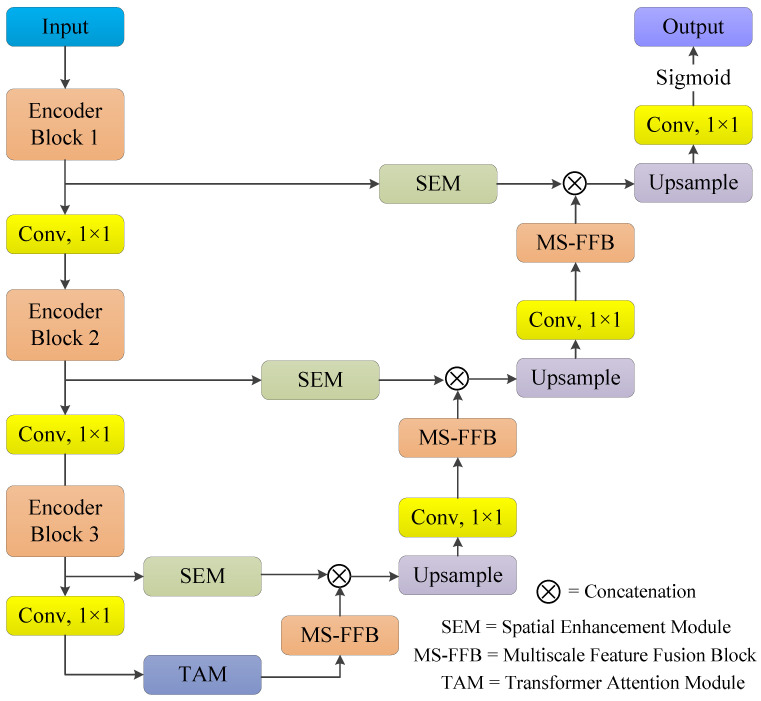
Architecture of the proposed CXR-Seg Network.

**Figure 2 bioengineering-12-00167-f002:**
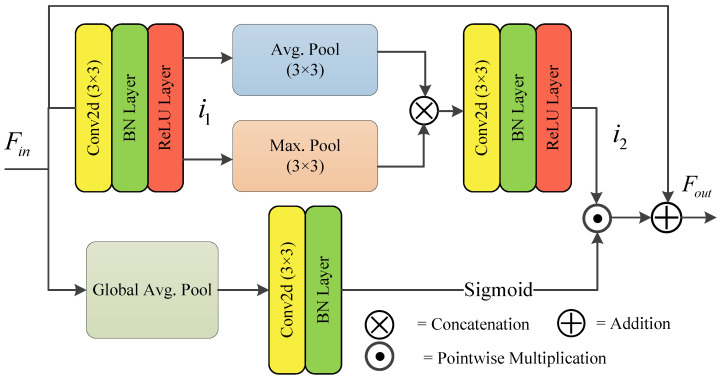
Structure of the spatial enhancement module (SEM) block.

**Figure 3 bioengineering-12-00167-f003:**
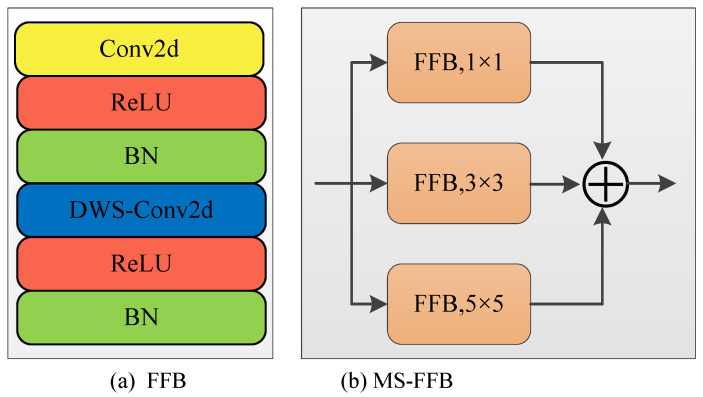
Schematics of the proposed multi-scale feature fusion block (MS-FFB): (**a**) feature fusion block, (**b**) multi-scale FFB.

**Figure 4 bioengineering-12-00167-f004:**
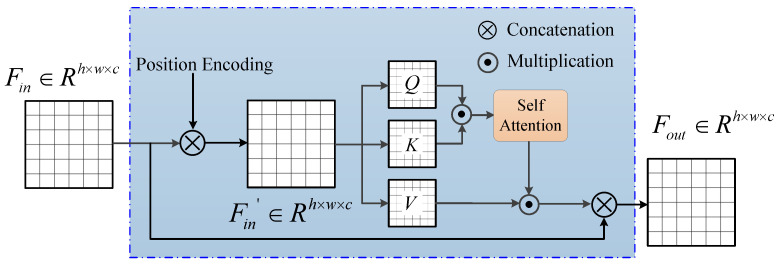
Schematic of the transformer attention module (TAM).

**Figure 5 bioengineering-12-00167-f005:**
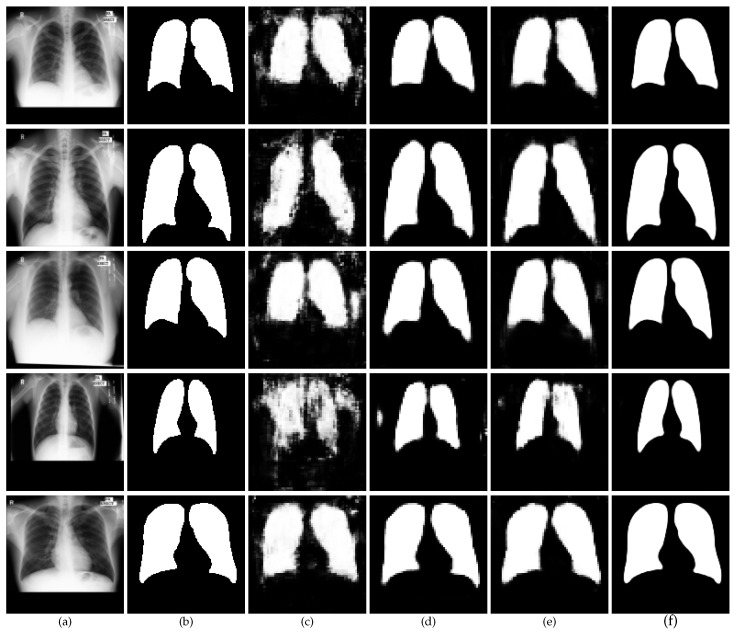
Visual performance of different network components of the proposed CXR-Seg Network on the MC dataset [30]. (**a**) Input CXR images; (**b**) corresponding ground truth images; (**c**) segmentation output of the baseline network; (**d**) segmentation output of the baseline network with TAM; (**e**) segmentation output of the baseline network with SEM; (**f**) segmentation output of the final CXR-Seg Network (Baseline + TAM + SEM).

**Figure 6 bioengineering-12-00167-f006:**
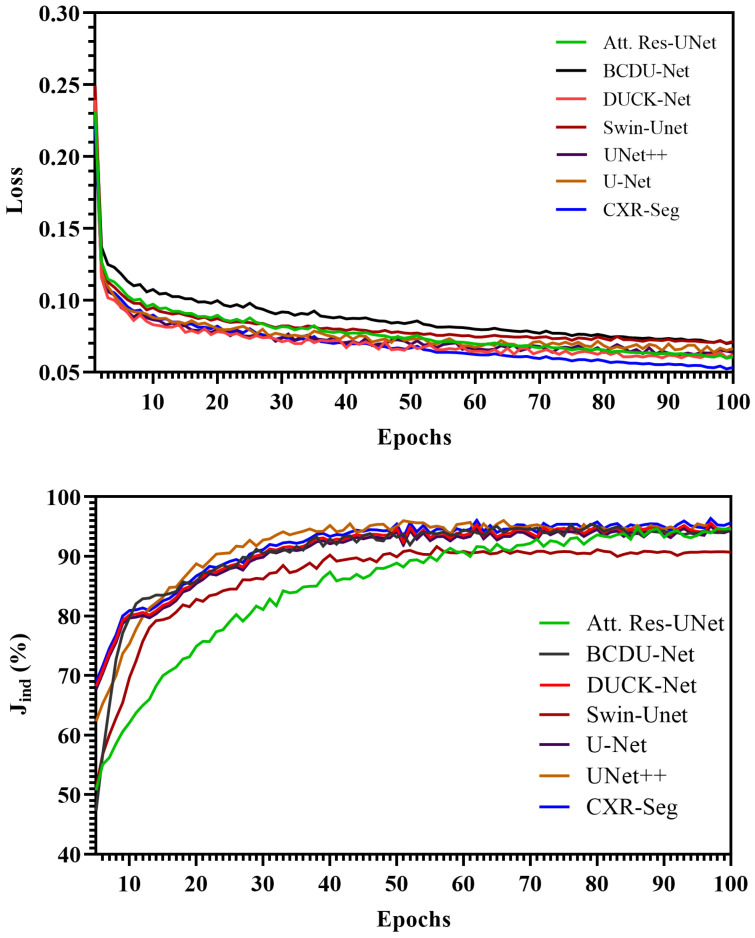
Performance of the proposed CXR-Seg Network with respect to the other methods in the MC dataset on the validation dataset. The first row illustrates the evolution of loss vs. epoch, while the second row depicts the Jaccard index vs. epoch behaviour.

**Figure 7 bioengineering-12-00167-f007:**
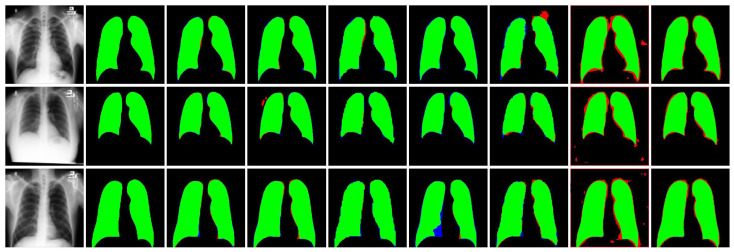
Visual performance comparison of the CXR-Seg Network with state-of-the-art methods on MC dataset [30]. From left to right: Input CXR images, corresponding ground truth images, segmentation output of CXR-Seg Network, Attention Res-UNet, BCDU-Net, DUCK-Net, SwinU-Net, U-Net, and U-Net++, respectively. True positives, false positives, false negatives and true negatives (pixels) are shown in green, red, blue, and black, respectively.

**Figure 8 bioengineering-12-00167-f008:**
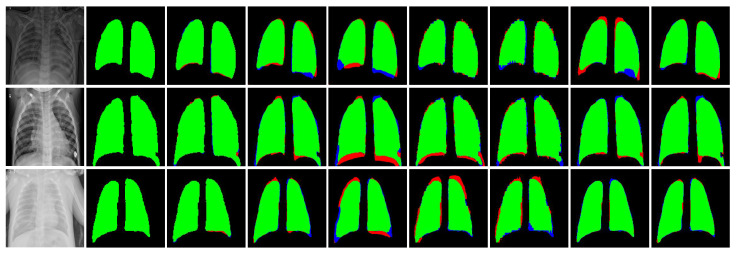
Visual performance comparison of the CXR-Seg Network with state-of-the-art methods on V7 Darwin COVID-19 Chest X-ray Dataset [31]. From left to right: Input CXR images, corresponding ground truth images, segmentation output of CXR-Seg Network, Attention Res-UNet, BCDU-Net, DUCK-Net, SwinU-Net, U-Net, and U-Net++, respectively. True positives, false positives, false negatives, and true negatives (pixels) are shown in green, red, blue, and black, respectively.

**Figure 9 bioengineering-12-00167-f009:**
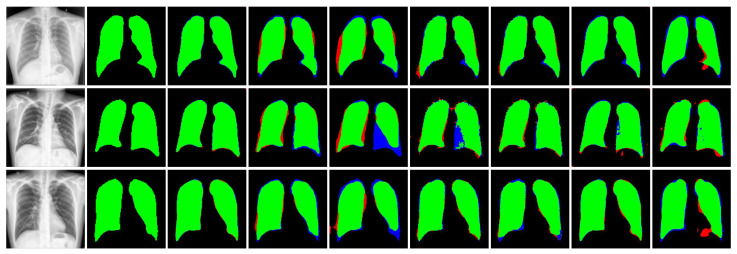
Visual performance comparison of the CXR-Seg Network with state-of-the-art methods on Shenzhen Tuberculosis CXR Dataset [30]. From left to right: Input CXR images, corresponding ground truth images, segmentation output of CXR-Seg Network, Attention Res-UNet, BCDU-Net, DUCK-Net, SwinU-Net, U-Net, and U-Net++, respectively. True positives, false positives, false negatives, and true negatives are shown in green, red, blue, and black, respectively.

**Figure 10 bioengineering-12-00167-f010:**
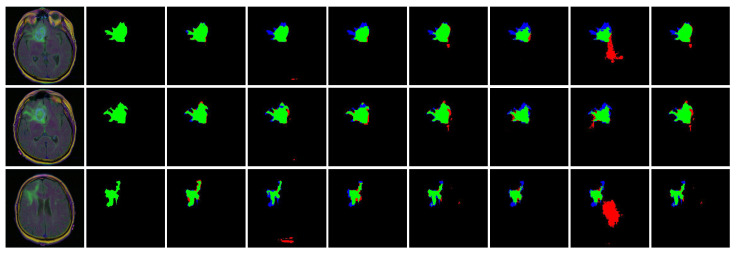
Visual performance comparison of the CXR-Seg Network with state-of-the-art methods on TCIA [32]. From left to right: Input MRI images, corresponding ground truth images, segmentation output of CXR-Seg Network, Attention Res-UNet, BCDU-Net, DUCK-Net, SwinU-Net, U-Net, and U-Net++, respectively. True positives, false positives, false negatives, and true negatives are shown in green, red, blue, and black, respectively.

**Table 1 bioengineering-12-00167-t001:** The datasets used for experimentation are described below. No augmentations were applied to the Darwin dataset due to its large size.

Dataset	Total Images	Image Resolution	Dataset Split			Augmentation			
			Training	Validation	Testing	Training	Validation	Test	Image Resize
			80%	10%	20%	Training×5	Validation×5	N.A	
MC [30]	138	4892×4020 to 4020×4892	110	14	14	550	70	14	256×256
Darwin [31]	6500	5600×4700 to 157×157	5200	650	650	5200	650	650	256×256
Shenzhen [30]	662	3000×3000	530	66	66	2650	330	66	256×256
TCIA	1084	256×256	868	108	108	4340	540	108	256×256

**Table 2 bioengineering-12-00167-t002:** Quantitative results of the ablation study performed on different components of the proposed CXR-Seg Network on the MC [30] dataset.

Method	Params (Millions)	Performance (%)				
		J	D	$A_{\mathrm{cc}}$	$S_{n}$	$S_{p}$
Baseline Network (BL)	2.08	81.47	89.67	94.52	89.21	96.39
Multi-scale Baseline Network (MBL)	3.72	85.47	91.70	95.20	92.01	97.30
MBL + TAM	4.39	90.81	95.15	97.42	95.37	98.09
MBL + SEM	4.94	91.52	95.55	97.60	95.56	98.30
CXR-Seg Network (MBL + TAM + SEM)	5.98	95.63	97.76	98.77	98.00	99.05

**Table 3 bioengineering-12-00167-t003:** Quantitative results of the ablation study performed on different backbones architectures for the proposed CXR-Seg Network on the MC [30] dataset.

Backbone	Performance (%)				
	J	D	$A_{\mathrm{cc}}$	$S_{n}$	$S_{p}$
ResNet	93.14	97.58	97.97	97.22	98.24
DenseNet	92.21	96.88	98.60	97.53	98.08
MobileNet	91.13	97.12	98.84	97.28	98.97
EfficientNet	95.63	97.76	98.77	98.00	99.05

**Table 4 bioengineering-12-00167-t004:** Performance comparison of the proposed model with various SOTA methods for chest X-ray image segmentation on the MC dataset.

Method	Performance (%)				
	J	D	$A_{\mathrm{cc}}$	$S_{n}$	$S_{p}$
Attention Res-UNet [34]	94.65	97.25	98.62	95.97	99.50
BCDU-Net [35]	94.53	97.19	98.57	96.00	99.43
DUCK-Net [36]	94.88	97.22	98.03	95.09	99.42
Swin-UNet [33]	90.69	95.07	97.61	94.34	98.65
U-Net [37]	94.38	97.10	98.53	95.99	99.40
UNet++ [38]	94.57	97.20	98.59	96.28	99.36
CXR-Seg Network	95.63	97.76	98.77	98.00	99.05

**Table 5 bioengineering-12-00167-t005:** Performance comparison of the proposed model with various SOTA methods for chest X-ray image segmentation on the V7 Darwin COVID-19 Chest X-ray Dataset.

Method	Performance (%)				
	J	D	$A_{\mathrm{cc}}$	$S_{n}$	$S_{p}$
Attention Res-UNet [34]	90.25	94.85	95.99	95.03	96.40
BCDU-Net [35]	89.42	94.42	95.79	94.68	96.24
DUCK-Net [36]	90.17	94.81	95.97	94.95	96.39
Swin-UNet [33]	89.92	94.68	95.85	95.11	96.22
U-Net [37]	89.83	94.63	95.89	94.86	96.32
UNet++ [12]	90.05	94.75	95.91	95.03	96.30
CXR-Seg Network	91.66	95.62	96.35	95.53	96.94

**Table 6 bioengineering-12-00167-t006:** Performance comparison of the proposed model with various SOTA methods for chest X-ray image segmentation on the Shenzhen Tuberculosis CXR Dataset.

Method	Performance (%)				
	J	D	$A_{\mathrm{cc}}$	$S_{n}$	$S_{p}$
Attention Res-UNet [34]	92.86	95.55	96.33	95.51	96.86
BCDU-Net [35]	88.22	95.12	96.13	95.17	96.70
DUCK-Net [36]	91.48	95.51	96.31	95.43	96.85
Swin-UNet [33]	91.23	95.38	96.19	95.59	96.68
U-Net [37]	91.14	95.34	96.23	95.34	96.78
UNet++ [12]	91.35	95.45	96.25	95.51	96.76
CXR-Seg Network	92.97	96.32	96.69	96.01	97.40

**Table 7 bioengineering-12-00167-t007:** Performance comparison of the proposed model with various SOTA methods on brain tumour segmentation using the TCIA dataset.

Method	Performance (%)				
	J	D	$A_{\mathrm{cc}}$	$S_{n}$	$S_{p}$
Attention Res-UNet [34]	81.68	86.73	98.67	85.81	99.35
BCDU-Net [35]	84.18	87.97	98.45	88.16	99.10
DUCK-Net [36]	84.33	87.32	98.87	88.12	99.01
Swin-UNet [33]	83.46	87.86	98.81	91.64	99.23
U-Net [37]	86.15	90.28	98.67	89.26	99.27
UNet++ [38]	78.44	83.42	98.22	85.69	98.88
CXR-Seg Network	92.93	95.47	99.34	95.63	99.79

**Table 8 bioengineering-12-00167-t008:** Computational complexity comparison of CXR-Seg Network with other methods.

Method	# of Params (M)	Memory Usage (MB)	Inference Time (ms)
Attention Res-UNet [34]	22.80	12.5	75.0
BCDU-Net [35]	13.30	8.0	60.0
DUCK-Net [36]	19.00	11.0	70.0
Swin-UNet [33]	26.00	14.0	80.0
U-Net [37]	7.00	5.5	50.0
UNet++ [38]	31.00	15.0	85.0
CXR-Seg Network	5.98	5.25	45.0

## Data Availability

The authors conducted experiments using three widely used datasets for lung segmentation (MC, Darwin, and Shenzhen) and brain tumour segmentation (TCIA) The performance of the multi-label classifier was evaluated on three publicly available CXR datasets: MC, JSRT, and NIH Chest X-ray.

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
