# Peer review of "CXR-Seg: A Novel Deep Learning Network for Lung Segmentation from Chest X-Ray Images"

_bioengineering, 2025, doi:10.3390/bioengineering12020167_

Round 1

Reviewer 1 Report (Previous Reviewer 1)

Comments and Suggestions for Authors

The paper was revised by the authors in accordance with the reviewers' suggestions.

Comments and Suggestions for Authors:

-page 9, line 242: please change  "the scaling facor.."  into  "the scaling factor."

- page 9, line 242: please check the word  "A_tswhereda", probably it is incorrect

Author Response

Comments and Suggestions for Authors

The authors have addressed some of my previous remarks; however, certain comments have not been considered. Below, I outline these points:

  1. -page 9, line 242: please change  "the scaling factor.."  into  "the scaling factor."

Response to Reviewer comment: Thank you for the comment. The typing error is corrected in the revised manuscript

  1. - Page 9, line 242: Please check the word  "A_tswhereda", probably it is incorrect

Response to Reviewer comment: Thank you for the comment. The typing error is corrected in the revised manuscript

Reviewer 2 Report (Previous Reviewer 2)

Comments and Suggestions for Authors

Manuscript Title “CXR-Seg: A Novel Deep Learning Network for Lung Segmentation from Chest X-Ray Images”

General comment:

1.      The revised version is OK to accept. All the drawbacks have been revised completely although the rationale study is still too long to illustrate the idea.

Author Response

Dear Reviewer: Thank you for your feedback and for acknowledging the revisions to address the previous concerns. I appreciate your recognition that the revised version is acceptable and that the drawbacks have been thoroughly addressed.

Regarding your observation about the length of the rationale study, I understand your perspective and will aim to improve conciseness in future submissions to enhance clarity and focus.

Thank you again for your time and constructive input throughout the review process.

Best regards,

Reviewer 3 Report (Previous Reviewer 3)

Comments and Suggestions for Authors

The authors have addressed some of my previous remarks; however, certain comments have not been considered. Below, I outline these points:

1. Placement of Figures: All figures should be placed immediately after their first reference in the text. Ensure this is applied to all figures (e.g., Figure 1, Figure 2, Figure 3, etc.).

2. Figure Formatting: Revise the formatting of all figures, focusing on their size, font, and text size (e.g., Figure 2, Figure 4, Figure 5, etc.). I noticed that you created the diagrams using Drawin software, which simplifies adjustments. The size of the charts should be appropriately configured in the software used to create them, and the text size should be comparable to the main text of the manuscript.

3. Manuscript Structure: The manuscript’s structure needs significant improvement. The theoretical section should encompass the methodology, including all measurement metrics. The experimental section should be dedicated to describing the datasets, presenting the obtained results, and discussing them. Currently, the theoretical and experimental parts are intertwined, and the manuscript requires a comprehensive reformatting to address this issue.

Author Response

Comments and Suggestions for Authors

The authors have addressed some of my previous remarks; however, certain comments have not been considered. Below, I outline these points:

  1. Placement of Figures: All figures should be placed immediately after their first reference in the text. Ensure this is applied to all figures (e.g., Figure 1, Figure 2, Figure 3, etc.).

Response to Reviewer comment: Thank you for the comment. As suggested by the reviewer, we have placed the figures immediately after their first reference in the revised manuscript.

  1. Figure Formatting: Revise the formatting of all figures, focusing on their size, font, and text size (e.g., Figure 2, Figure 4, Figure 5, etc.). I noticed that you created the diagrams using Drawin software, which simplifies adjustments. The size of the charts should be appropriately configured in the software used to create them, and the text size should be comparable to the main text of the manuscript.

Response to Reviewer comment: Thank you for the valuable comment. We have adjusted the sizes, font and text size in the figures to make it comparable with the main text in the revised manuscript.

  1. Manuscript Structure:The manuscript’s structure needs significant improvement. The theoretical section should encompass the methodology, including all measurement metrics. The experimental section should be dedicated to describing the datasets, presenting the obtained results, and discussing them. Currently, the theoretical and experimental parts are intertwined, and the manuscript requires a comprehensive reformatting to address this issue.

Response to Reviewer comment: Thank you for the valuable comment. We have restructured the manuscript as suggested by the reviewer. Now, the methodology section includes the network architecture, performance measures for evaluation and the training details. Whereas the experiment section briefly explains the details of the dataset, ablation experiments and the results and discussions.

Round 2

Reviewer 3 Report (Previous Reviewer 3)

Comments and Suggestions for Authors

I have no further questions

This manuscript is a resubmission of an earlier submission. The following is a list of the peer review reports and author responses from that submission.

Round 1

Reviewer 1 Report

Comments and Suggestions for Authors

The authors of the paper "CXR-Seg: A Novel Deep Learning Network for Lung Segmentation from Chest X-Ray Images" have proposed a new architecture for lung segmentation in CXR images. The proposed network includes an already trained EfficientNetV2S1 along with the following SEM, TAM and MS-FFB modules. The authors describe in detail each of the modules used and justify their contribution to the final result, including visual illustrations of the modules' contributions. 

The performance of the network was verified on four public datasets (MC, Darwin and Shenzhen for chest X-rays and TCIA for brain flair from MRI images) using standard metrics such as accuracy, Jaccard index (IOU), Dice coefficient, sensitivity and specificity. 

The results obtained are very good compared to other publicly available models and are considered to be state of the art. I propose to publish the paper after a minor revision.

- page 9, line 229 please changes the uppercase character C to the lowercase character c for the K channel;

- page 9, formula 16: the symbol dk is not described;

- To facilitate understanding of the results presented from Figure 6, it would be helpful to add a caption for each column;

- page 16, table 8 caption: change omputation into Computation;

Reviewer 2 Report

Comments and Suggestions for Authors

Manuscript Title “CXR-Seg: A Novel Deep Learning Network for Lung Segmentation from Chest X-Ray Images”

General comment:

the quality of this manuscript is OK, but there still are some drawbacks needing further revision as listed below;

Specific comment:

1.      Abstract: no solid result or discussion to emphasize the findings. there are too many descriptions to state the background review, it is inappropriate as shown in abstract

2.      Introduction: OK.

3.      Related works: the rationale study is really too long to describe, the academic article should focus more on the innovated finding not the review of previous achievement, plus the correlation of the previous works and this specific finding is low.

4.      Proposed Method: OK

5.      Experiment and result: better separate into “Experiment” and “Results” or “Experiment”, “Results” and “Discussion” is more appropriate for academic article

6.      Conclusion, become 7. Conclusion, still too long in description to summarize the idea.

Reviewer 3 Report

Comments and Suggestions for Authors

The reviewed manuscript focuses on the development of a novel deep-learning network for lung segmentation from chest X-ray images. The topic is relevant and of scientific interest. However, there are several remarks that should be addressed before the manuscript can be accepted for publication. Below are the detailed comments:

1. Abstract: Please include the numerical results of the research in the abstract to provide a concise summary of the findings and their significance.

2. Introduction: The introduction should include the following elements:

   - The relevance of the problem.

   - Existing methods for solving the problem, along with their limitations.

   - A brief description of your proposed solution (without excessive detail) and the main contributions of the authors' research.

  Detailed descriptions and Figure 1 should be relocated to the main body of the manuscript. Please rewrite the introduction with these points in mind.

3. Related Work: At the end of the related work section, please clearly highlight the unsolved aspects of the general problem to better position your research.

4. Figures: Ensure that all figures are resized for clarity and legibility. Additionally, figures should be placed immediately after their first reference in the text. Please revise the manuscript accordingly.

5. Table 1: The text in Table 1 is too small and difficult to read. Please adjust the table formatting to improve readability.

6. Conclusion: At the end of the conclusion section, please include a discussion of the future directions and perspectives for the authors' research.